# In-Cell Determination of Lactate Dehydrogenase Activity in a Luminal Breast Cancer Model – *ex vivo* Investigation of Excised Xenograft Tumor Slices Using dDNP Hyperpolarized [1-^13^C]pyruvate

**DOI:** 10.3390/s19092089

**Published:** 2019-05-05

**Authors:** Yael Adler-Levy, Atara Nardi-Schreiber, Talia Harris, David Shaul, Sivaranjan Uppala, Gal Sapir, Naama Lev-Cohain, Jacob Sosna, Shraga Nahum Goldberg, J. Moshe Gomori, Rachel Katz-Brull

**Affiliations:** Department of Radiology, Hadassah Medical Center, Hebrew University of Jerusalem, The Faculty of Medicine, Jerusalem 9112001, Israel; ayal@hadassah.org.il (Y.A.-L.); atara.nardi@mail.huji.ac.il (A.N.-S.); talia.harris@mail.huji.ac.il (T.H.); david.shaul@mail.huji.ac.il (D.S.); uppala.sivaranjan@mail.huji.ac.il (S.U.); gal.sapir1@mail.huji.ac.il (G.S.); naamal@hadassah.org.il (N.L.-C.); jacobs@hadassah.org.il (J.S.); snahum@hadassah.org.il (S.N.G.); gomori@hadassah.org.il (J.M.G.)

**Keywords:** LDH, ^13^C-NMR, hyperpolarization, magnetic resonance, breast cancer, precision-cut tissue slices

## Abstract

[1-^13^C]pyruvate, the most widely used compound in dissolution-dynamic nuclear polarization (dDNP) magnetic resonance (MR), enables the visualization of lactate dehydrogenase (LDH) activity. This activity had been demonstrated in a wide variety of cancer models, ranging from cultured cells, to xenograft models, to human tumors in situ. Here we quantified the LDH activity in precision cut tumor slices (PCTS) of breast cancer xenografts. The Michigan Cancer Foundation-7 (MCF7) cell-line was chosen as a model for the luminal breast cancer type which is hormone responsive and is highly prevalent. The LDH activity, which was manifested as [1-^13^C]lactate production in the tumor slices, ranged between 3.8 and 6.1 nmole/nmole adenosine tri-phosphate (ATP) in 1 min (average 4.6 ± 1.0) on three different experimental set-ups consisting of arrested vs. continuous perfusion and non-selective and selective RF pulsation schemes and combinations thereof. This rate was converted to an expected LDH activity in a mass ranging between 3.3 and 5.2 µmole/g in 1 min, using the ATP level of these tumors. This indicated the likely utility of this approach in clinical dDNP of the human breast and may be useful as guidance for treatment response assessment in a large number of tumor types and therapies ex vivo.

## 1. Introduction

It is recognized that the varying clinicopathological phenotype of breast malignancy closely parallels at least four intrinsic molecular subtypes: luminal A, luminal B, HER2 positive with estrogen receptor negative disease, and triple negative disease [1]. These parameters influence treatment choices, such as anti-hormonal therapy for luminal A and B or specific antibody treatment for Her 2 positive cancers. Luminal disease is defined as estrogen positive disease. Further subdivision of this subtype to A and B is largely based on tumor biology as currently defined by immunohistochemistry, although multigene testing plays a role in categorizing the two as well [2]. These characterizations further influence treatment choices, as it is in these patients that uncertainty about optimal treatment most commonly arises, as clinicians seek to avoid over- or under-treatment [3].

Although significant advancement has been made in medical imaging in optimizing the diagnostic performance and lesion characterization in conventional morphology based modalities, such as mammography and ultrasound (US), even imaging incorporating pathophysiological parameters such as contrast enhanced breast MRI or diffusion weighted imaging are still limited in the ability to guide such treatment choices in breast cancer and in general in making a differential diagnosis of breast cancer [4,5,6,7,8,9,10]. Specifically, certain imaging findings characterize the luminal subtype: for example, on mammograms, architectural distortion and spiculated mass are often seen; and on US, hallmarks of this subtype consist of posterior acoustic shadowing, irregular shape, angular or spiculated margin, and echogenic halo [11,12,13,14]. However, in general in breast cancer, overlap in anatomic imaging findings, and the significant implications in deferring the diagnosis, often require definitive pathological evaluation, which more often than not, will result in benign findings [15,16,17,18].

The dissolution-dynamic nuclear polarization (dDNP) hyperpolarized MRI technology developed by Ardenkjaer-Larsen et al. [19] is growing in use in clinical studies [20,21,22,23,24,25] mostly focusing on cancer tissues [20,22,23,25]. As opposed to conventional carbon-13 nuclear magnetic resonance spectroscopy (^13^C-NMR), the dDNP technology enhances the ^13^C-NMR signal of a ^13^C labeled compound about 10,000 fold, such that in-cell metabolism can be monitored at very high temporal resolution (seconds) and at concentrations that are physiologically relevant. In all of the clinical dDNP studies [20,21,22] performed to date, the metabolism of [1-^13^C]pyruvate was suggested as a promising tool for differential diagnosis on medical MRI as well as it being the only dDNP molecular probe currently approved for human use.

In preclinical studies, hyperpolarized [1-^13^C]pyruvate metabolism has been utilized for studying cancer metabolism in cultured cells [26,27,28,29,30] and in vivo – both in xenograft tumors [31,32,33,34,35] and in transgenic cancer models in animals [36,37]. Specifically, with respect to breast cancer a few studies were performed in cultured cells [26,27,30] and one study was performed in vivo [35]. All of the dDNP studies of cancer (clinical and pre-clinical) using [1-^13^C]pyruvate have demonstrated the production of hyperpolarized [1-^13^C]lactate in the cells or tumors, indicating the activity of lactate dehydrogenase (LDH) and utilizing it as an indicator of micro-environmental changes and of response to treatment [28,30,34,35,38,39]. These findings are in agreement with LDH activity determined in malignant tissues using biochemical assays which have unanimously demonstrated elevated LDH levels or activities in cancer [40,41,42,43]. 

The advantage of the dDNP-MRI technology is in its non-invasive and non-destructive nature, which potentially will be able to make use of such robust cellular characteristics without penetrating the body and without ionizing radiation. To the best of our knowledge, clinical dDNP studies of breast cancer are yet to be performed. Considering the prevalence of the disease and the complex nature of such clinical studies, preclinical guidance towards such potential studies is warranted. In addition, the ability to determine in-cell metabolism ex vivo non-destructively has advantages over *in vivo* metabolic imaging as the effect of therapeutic treatments may be evaluated on patient derived tumor slices and the best treatment course can be chosen based on those measurements. Our aim in the current study was to perform a preclinical evaluation of the activity of LDH in a luminal breast cancer model, as demonstrated by dDNP hyperpolarized [1-^13^C]pyruvate metabolism. We have applied several methodologies for assessing this activity and quantifying it. In this way the potential visibility of this metabolic activity in human breast cancer in vivo and *ex vivo* was evaluated. 

Xenograft tumors of the human breast cancer veteran MCF7 cell-line were chosen here as a model representing the luminal human breast cancer subgroup due to its expression of the Estrogen receptor [44]. Because metabolism in other bodily tissues of the hosting animal can influence the metabolism seen in the target tissue, as was recently demonstrated in the rat heart [45], we have devised a methodology to observe the metabolism in the tumor exclusively. To this end, we have developed a new model consisting of viable precision-cut tissue slices of a xenograft tumor (implanted in a mouse). Such tumor slices allow the interrogation of the tumor metabolism exclusively on one hand, and on the other hand provide a more complete model of the tumor than cultured cells because they represent the 3D tumor tissue architecture and involve parenchymal cells, stromal cells, vasculature structure (even if cut and not active), extracellular matrix, and cell-cell interactions. 

dDNP-MR studies of perfused whole organs have been performed previously in the rodent heart [46,47,48,49] and liver [50]. With regard to tissues that cannot be perfused ex vivo via their own vasculature, we have recently developed a dDNP-MR set-up for precision cut tissue slices [51,52]. This set-up was recently applied for monitoring of hyperpolarized [1-^13^C]pyruvate metabolism in the rodent brain and liver [51,52], whereas for the former, a perfused whole organ system is not feasible and for the latter, the rodent precision cut tissue slices model provided a baseline for future studies in such slices that may be produced from small pieces of the human liver obtained during a required medical procedure (and therefore cannot be perfused via own vasculature). Both arguments are also valid for performing ex vivo studies on xenograft tumors using the precision cut tissue approach: (1) an ex vivo system for perfusion of a whole tumor is not feasible, and (2) such studies can form a baseline for studies on precision-cut tissue slices of human breast lesions obtained in the course of a medical procedure, as guidance for clinical in vivo dDNP-MRI studies, and for treatment response assessment in a large number of tumor types and therapies ex vivo. We note that cultured prostate tumor slices from human origin [53] and patient derived kidney tumor slices [54] have been previously investigated using hyperpolarized [1-^13^C]pyruvate and demonstrated [1-^13^C]lactate production and diagnostic capability.

The current study was therefore designed to serve the following aims (1) To study the metabolism of hyperpolarized [1-^13^C]pyruvate in a well-established hormone-responsive breast cancer tumor model within a 3D architecture and intact inter-cell and extracellular interactions; (2) To quantify this metabolism without wash-in of metabolites from other body organs; (3) To set a baseline for *ex vivo* studies of breast cancer tumors derived from multiple breast cancer cell lines of varying aggressiveness levels [44] and for patient derived xenografts [55]; (4) To set a baseline for studies of *ex vivo* breast tissues obtained from patients in real time; and 5) To provide guidelines for the feasibility of non-invasive study of hyperpolarized [1-^13^C]pyruvate metabolism in the clinical setting.

## 2. Materials and Methods

### 2.1. Chemicals

The OXO63 radical (GE Healthcare, Little Chalfont, UK) was obtained from Oxford Instruments Molecular Biotools (Oxford, UK). [1-^13^C]pyruvic acid was purchased from Sigma-Aldrich, (Rehovot, Israel) and from Cambridge Isotope Laboratories (Tewksbury, MA, USA). Dulbecco’s Modified Eagle Medium (DMEM) without glutamine, glucose, and sodium pyruvate, and L-glutamine were purchased from Biological Industries (Beit Ha’Emek, Israel). D-Glucose was purchased from Sigma-Aldrich. Matrigel matrix (5 mL/case) was purchased from Bactlab (Cat. No FAL356234, Caesarea, Israel). Ketamine solution (100 mg/mL), xylazine solution (23.32 mg/mL), and isoflurane were obtained from the Institutional Authority for Biological and Biomedical Models of the Hebrew University.

### 2.2. Animals 

All of the experimental procedures were in accord with the regulations of the institutional animal care and use committee of the Hebrew University (MD-17-15048-5). Care was taken to minimize pain and discomfort to the animals. Eleven C.B-17/IcrHsd-Prkdc-SCID-Lyst-bg female mice (Harlan, Rehovot, Israel), aged 7–8 weeks were injected with a solution of 1-4 million MCF7 human breast cancer cells (100 µL) combined with 100 µL of Matrigel in total injected volume of 200 µL per mouse. After 2–6 months, the tumor size reached a final volume of 1–1.2 cm^3^ (estimated by a caliper), after which, the tumors were extracted and sliced as described below. The number of tumors per animal and their utilization is summarized in the Appendix A.

### 2.3. Surgical Procedure and Slice Preparation

All of the surgical procedures began after placing the animals under general anesthesia and obtaining a negative pedal pain reflex. Anesthesia was performed with either (1) an intraperitoneal injection of ketamine and xylazine mixture (3:2 v/v at a dose of 0.12 mL mixture per 100 g body) or (2) with isoflurane using a gas anesthesia system (Somnosuite, Kent Scientific, Torrington, CT, USA), using 3.5% isoflurane for induction and 3% isoflurane for maintenance and 340 mL/min air flow.

Following a subcutaneous incision, the tumor was exposed, rapidly resected, and placed in ice-cold (1–2 °C) DMEM medium. The animals were then immediately sacrificed by cervical dislocation. Whole tumor precision cut tissue slices (500 µm) were prepared using a McIlwain tissue chopper (The Mickle Laboratory Engineering Company Ltd., Surrey, UK). The process of tumor extraction and slicing took approximately 30 min. Throughout this time the tissue was constantly kept at an ice-cold temperature DMEM bubbled with 95%/5% O_2_/CO_2_. This process is illustrated in Figure 1.

### 2.4. Perfusion Media and the Perfusion System 

MCF7 tumor slices were perfused with DMEM medium. The medium was supplemented with 25 mM D-glucose, 2 mM L-glutamine, and 5 % D_2_O on the day of the experiment. The medium was bubbled with 95%/5% O_2_/CO_2_ for 1 h prior to slice perfusion and continuously bubbled with this gas mixture throughout the experiment at a flow rate 0.4 L/min. The pH of the medium was 7.4. A 100 mL reservoir of this medium was kept in a water bath at 40 °C outside the NMR spectrometer. This medium was delivered to the NMR tube containing the tissue slices (Figure 1). The perfusion system was made of medical grade extension tubes and the medium was circulated with a peristaltic pump (Masterflex L/S Analog Pump Systems, Cole-Parmer, Vernon Hills, IL, USA). Thin polyether ether ketone (PEEK) lines (id. 0.040”, Upchurch Scientific, Inc., Oak Harbor, WA, USA) were used for medium and hyperpolarized agent flow to and from the NMR tube. The magnetic susceptibility of PEEK is similar to water and therefore can be used during NMR spectroscopy recordings. The temperature of the slices inside the NMR spectrometer during the perfusion was 32–36 °C and was measured with an NMR compatible temperature probe (Osensa, Burnaby, BC, Canada).

### 2.5. Spin Polarization and Dissolution

Spin polarization of [1-^13^C]pyruvic acid and fast dissolution were carried out in a dDNP spin polarization device (HyperSense, Oxford Instruments Molecular Biotools, Oxford, UK) operating at 3.35 T. Microwave frequency of 94.116 GHz at 100 mW was applied for polarization of [1-^13^C]pyruvic acid at 1.4–1.49 K. The formulation consisted of 11.1–15 mM trityl radical (OX063) in the neat acid. The dissolution medium consisted of 50 mM phosphate buffer which contained 19 mM TRIS and 138.6 mM NaCl. The pH of the dissolution medium was adjusted with NaOH such that upon mixing with the pyruvic acid in the cup the final pH of the solution injected to the slices was 7.4. The solid hyperpolarized sample was quickly dissolved in 4 mL of the hot and pressurized dissolution medium (170 °C and 10 bar). This was injected through a Teflon line of about 2.4 m length with 6 s of He(g) chase from the spin-polarizer into a conical tube placed at the fringe field of the magnet. For volumes larger than 4 mL, additional phosphate buffer medium was added to this conical tube prior to dissolution. The amount of [1-^13^C]pyruvic acid formulation placed in the polarization cup was 5 ± 0.6 mg, per 4 mL of hyperpolarized medium injected to the NMR tube.

### 2.6. Experimental Design: Hyperpolarized Media Injection and Acquisition Approach

The introduction of the hyperpolarized solution to the slices, as well as the data acquisition that followed, were performed in three different approaches as summarized below:(1)*Manual bolus injection during perfusion arrest and non-selective RF pulses:* This is the most common hyperpolarized medium administration method and data acquisition approach in dDNP studies carried out in NMR tubes [27,29,30,51,52,54]. In this approach, the hyperpolarized solution was manually transferred via a manifold (connected to the in-flow line and made of a combination of medical grade 3-way valves and syringes) from the conical tube directly to the bottom of the NMR tube containing the tumor slices as previously described [52]. Altogether, the duration of hyperpolarized media transfer in this system was completed within 15 s from the start of the dissolution process. The hyperpolarized medium was injected gently to minimize tissue displacement and care was taken to avoid the introduction of air bubbles that could interfere with magnetic field homogeneity (due to the large difference in magnetic susceptibility between air and water). In this setup, the perfusion was stopped ~30 s before the injection of the hyperpolarized solution and was resumed only after the acquisition of the hyperpolarized spectra was completed (maximum of 4 min). This was done in order to characterize the metabolism of a constant concentration of [1-^13^C]pyruvate without the effects of wash-in and wash-out of the hyperpolarized medium.(2)*Manual bolus injection during perfusion arrest and selective RF pulses:* here, the introduction of the hyperpolarized solution was the same as in approach 1. However, the acquisition was performed using hyperpolarized product-selective saturating RF excitations, termed hereafter selective excitations, which fully excited the metabolite of interest while the precursor (pyruvate) is excited to a much lower degree. Therefore, only newly synthetized metabolites were detected in the consecutive excitation (see the Appendix A). A series of at least 4 such selective excitations was performed immediately after the end of the bolus injection with a repetition time of 1 to 5 s. This was done to quench the signals from [1-^13^C]pyruvate impurities that resonate close to the metabolite signals [56]. Then, the same pulses were applied with a repetition time of 8 to 16 s to record the metabolism.(3)*Administration of the hyperpolarized medium in a continuous flow and selective RF pulses:* in this approach, the hyperpolarized agent was introduced to the tissue via a bypass system. This involved combining the 4 mL of dissolution medium with 8 mL of heated and oxygenated phosphate buffer awaiting in the conical tube used for hyperpolarized media collection (described above). Then, these 12 mL of hyperpolarized medium were loaded into a heated extension tube (bypass line) connected to the perfusion system with a manifold made out of medical grade 3-way valves. Then, the content of the bypass line (12 mL) was infused into the NMR tube containing the slices at a rate of 4 mL/min using the perfusion system. The purpose of the bypass system was to ensure continuous delivery of well-oxygenated hyperpolarized medium throughout the metabolic investigation. An additional benefit of this perfusion system was avoiding turbulence and tissue movement that could be caused by the bolus injection. In this approach, acquisition was started following the loading of the hyperpolarized medium into the bypass line (prior to arrival of the hyperpolarized medium to the slices).

### 2.7. NMR Spectroscopy

^31^P- and ^13^C-NMR spectroscopy were performed in a 5.8 T non-shielded high resolution NMR spectrometer (RS2D, Mundolsheim, France) located about 2.2 m away from the spin-polarization magnet (center-to-center), using a 10 mm broad-band NMR probe. Homogeneity optimization (shim) was performed using the water signal on the ^1^H channel and using the lock system. To support the latter, the medium was supplemented with 5% D_2_O in all of the experiments.

#### 2.7.1. ^13^C-NMR Acquisitions with Non-selective RF Pulses

Non-selective (hard) pulses for ^13^C spectra acquisition were applied with a 12° nutation angle with a repetition time of 5 s. The bandwidth was selected to detect the entire spectral width of the ^13^C spectrum (300 ppm) enabling simultaneous detection of hyperpolarized [1-^13^C]pyruvate and its metabolites. The ^13^C-NMR acquisition was started immediately at the start of the dissolution process in the spin-polarizer, prior to the injection of the medium to the slices, such that the first acquisitions showed only noise and then the entrance of the hyperpolarized media into the tube and subsequent metabolism were recorded in real-time.

#### 2.7.2. ^13^C-NMR Acquisitions with Selective RF Pulses

Selective excitations for ^13^C acquisitions (soft pulses) were given with a ~90° nutation angle using a 2.5 ms cardinal sine (sinc) pulse and excited the [1-^13^C]lactate signal as described above and in the Appendix A. For determination of metabolic rates, selective pulses were acquired at a repetition time (TR) of 8 or 12 or 16 s. The pulse was centered either at the [1-^13^C]pyruvate hydrate resonance (179.8 ppm) or at 186.6 ppm (both are 214 Hz off-resonance to the [1-^13^C]lactate signal), thereby applying full excitation (and depolarization) of the [1-^13^C]lactate signal and a very low excitation of the [1-^13^C]pyruvate signal (lower in the latter frequency, Appendix A).

#### 2.7.3. ^31^P-NMR Spectroscopy

^31^P spectra of thermal equilibrium phosphate signals were acquired with a nutation angle of 50° and a repetition time of 1.1 s. The bandwidth used (~79 ppm) was larger than the span required to detect the high energy phosphates (~40 ppm) due to technical limitations of the instrument. This introduced about 20% increase in the noise level. 

#### 2.7.4. Alternating ^31^P- and ^13^C-NMR Acquisitions

The probe was tuned back and forth from ^31^P to ^13^C during the experiment to support the requirement of the experimental workflow, in which several injections of hyperpolarized media were administered to the same perfused tumor slices sample – requiring ^13^C acquisitions, and the energy status was monitored using ^31^P spectroscopy before, in between, and after the injections. 

### 2.8. Validation of Viability

To monitor the viability status of the slices within the spectrometer throughout the study, we used ^31^P-NMR spectroscopy. The ^31^P-NMR spectra showed the signals of α, β, and γ ^31^P nuclei of adenosine triphosphate (ATP), the ^31^P nucleus of inorganic phosphate (Pi), and signals due to phosphomonoesters (PME) and phosphodiesters (PDE).

### 2.9. Processing and Data Analysis

#### 2.9.1. Software

Spectral processing was performed using MNova (Mestrelab Research, Santiago de Compostela, Spain). Integrated intensities were calculated either with MNova or with DMFIT [57]. Curve fitting was performed using Matlab (Mathworks, Natick, MA, USA).

#### 2.9.2. Determination of [1-^13^C]pyruvate T_1_ and Utilization of This Value

In experiments with perfusion arrest and selective pulse it was possible to determine the T_1_ of [1-^13^C]pyruvate using a curve fitting of the signal decay to Equation (1):(1)$$M\left(t\right)=M_{o}\cdot e^{\left(\frac{-t}{T_{1}}\right)}\cdot \cos\theta^{\left(\frac{t}{TR}\right)}$$
in which *TR*, the time between excitations, and *θ*, the nutation angle of excitation, are known. Equation (1) reflects the decay of the signal due to T_1_ and RF pulsation only because [1-^13^C]pyruvate was given in excess and the small amount that was metabolized appeared negligible. The average T_1_ of [1-^13^C]pyruvate in these experiments was found to be 53.4 ± 5.8 s (n = 5). This T_1_ value was then used in the analysis of the experiments with continuous flow as described below. 

The perfusion was turned back on only after the hyperpolarized [1-^13^C]pyruvate signal decayed to ensure that the decay curve reflects only response to the RF pulses and T_1_ decay and is not affected by wash-out of the hyperpolarized material out of the NMR tube.

#### 2.9.3. T_1_ Correction of the Hyperpolarized [1-^13^C]pyruvate Signal and Characterization of [1-^13^C]pyruvate Concentration in the Medium 

To determine the metabolic rates in the experiments, the [1-^13^C]pyruvate concentration in the medium surrounding the slices (and visible to the probe) must be known. In the experiments with arrested perfusion this concentration is known as the hyperpolarized medium completely replaces the medium in the NMR probe. However, in the experiments with continuous flow, the concentration of hyperpolarized [1-^13^C]pyruvate is not simple to assess as it depends on the exact timing of arrival to the NMR tube and the signal itself is affected by T_1_ decay. Nevertheless, the concentration of hyperpolarized pyruvate in the bypass line is known and constant. Thus, we needed to find the time points when the concentration is constant within the volume visible to the NMR probe. To identify these time points, we applied a correction based on T_1_ to the hyperpolarized [1-^13^C]pyruvate data according to Equation (2):(2)$$S_{Pyr\_corr}=\frac{S_{Pyr}}{e^{\left(\frac{-t}{T_{1}}\right)}}$$
where $S_{\mathrm{Pyr}\_\mathrm{corr}}$, is the corrected [1-^13^C]pyruvate signal, $S_{\mathrm{Pyr}}$ is the signal intensity of the [1-^13^C]pyruvate signal at a given time, t, decaying according to the T_1_ that was determined as described above in the arrested perfusion experiments. The time points which were within 10% difference from the first maximum point of this corrected signal were determined to represent the duration in which the [1-^13^C]pyruvate concentration was constant. 

#### 2.9.4. ATP Concentration 

ATP concentration was calculated from the γ-ATP signal of the tumor slices on the thermal equilibrium ^31^P spectra and an ATP standard sample acquired on the same experimental day (111 mM in D_2_O). The differences in flip angle, number of acquisitions, and relaxation effects were taken into account. The amount of ATP in the samples (in the volume detected by the probe) ranged between 4.1 to 22.3 nmole and averaged at 11.4 ± 6.1 nmole.

#### 2.9.5. Metabolic Rate Calculation

For the analysis of rate constants for the enzymatic conversions, the tissue slices volume and the medium volume observed by the NMR probe were estimated to be about 0.9 mL and 0.5 mL, respectively, whereas the total volume observed by the probe is 1.4 mL. The enzymatic rates from the experiments with non-selective excitation pulses were calculated based on a kinetic model previously developed in our lab [58]. The enzymatic rates from the experiments with the selective excitation pulses were calculated as follows. The rate of hyperpolarized [1-^13^C]lactate production, ν, per nmole ATP of the tissue, was calculated from its signal intensity, S_lac_, relative to that of pyruvate, S_pyr_, taking into account (1) the concentration of hyperpolarized [1-^13^C]pyruvate, [Pyr] (14 mM); (2) the ratio of [1-^13^C]lactate and [1-^13^C]pyruvate relative response to the selective excitation pulse (ρ) (Appendix A); (3) the repetition time (TR); (4) the volume of the hyperpolarized medium, V_medium_ (0.5 mL); and (5) the ATP content of the tissue observed by the probe. This calculation is summarized in Equation (3):(3)$$v=\frac{S_{\mathrm{lac}}}{S_{\mathrm{pyr}}}\cdot \frac{\left[\mathrm{Pyr}\right]\cdot \rho }{\mathrm{TR}}\cdot \frac{\left(V_{\mathrm{medium}}\right)}{\mathrm{nmol}\;\mathrm{ATP}}$$

## 3. Results

We were able to maintain the viability of the tumor slices for more than 7 h under controlled perfusion conditions. Figure 2 shows ^31^P spectra acquired at thermal equilibrium at 1.5, 5.5, and 7.5 h of perfusion within the spectrometer. Within this time frame, consecutive injections of the hyperpolarized metabolic agent were typically performed. The spectra show signals due to the three ^31^P nuclei of adenosine triphosphate (ATP), and due to the ^31^P nuclei of inorganic phosphate (Pi), phosphomonoesters (PME), and phosphodiesters (PDE). In all of the experiments described here, the levels of ATP were the same before and after hyperpolarized media injections, which suggested a stable energy status of the tissue.

After validating the viability of the tumor slices perfused in the NMR spectrometer we proceeded to test the metabolism of hyperpolarized [1-^13^C]pyruvate in the slices. Typical spectra obtained using non-selective RF excitation and arrested perfusion (Experimental design 1, Methods) are shown in Figure 3. Following the injection of hyperpolarized [1-^13^C]pyruvate, the build-up and later the decay of hyperpolarized [1-^13^C]lactate was observed at 183.2 ppm. The signals of [1-^13^C]pyruvate and its hydrate form were observed at 171.0 and 179.4 ppm, respectively.

To determine the rate of [1-^13^C]lactate production in the tumor slices in this experimental design we applied a kinetic model which takes into account a production via first order reaction and decay of the hyperpolarized state of both the substrate and the product due to T_1_ relaxation and RF pulsation, as previously described [58]. Figure 4 demonstrates such a typical fit of the experimental data to the kinetic model.

Altogether three such studies were performed on two different tissue samples (i.e. on two different experimental days), whereas each sample contained tumor slices obtained from 3 different animals (Appendix A). The average rate constant of [1-^13^C]lactate production in the tumor slices (*k_1st_)* was found to be 1.67 ± 0.13 x 10^-4^ s^−1^. The average T_1_ for [1-^13^C]pyruvate was 58.3 ± 7.6 s and the average T_1_ for [1-^13^C]lactate was 19.3 ± 2.4 s. The R^2^ of the fits for [1-^13^C]pyruvate ranged from 0.998 to 1. The R^2^ of the fits for [1-^13^C]lactate ranged from 0.75 to 0.99.

To use these values to predict the amount of [1-^13^C]lactate that would be produced at a given time by breast cancer tissue we used Equation (4):(4)$$A_{lac}=Pyr\cdot \left(1-e^{\left(-k_{1\mathrm{st}}\cdot t\right)}\right)$$
where *A_lac_* is the amount of [1-^13^C]lactate produced during time, *t*, in the presence of [1-^13^C]pyruvate concertation [*Pyr*]. *A_lac_* is calculated with reference to the total [1-^13^C]pyruvate amount, *Pyr*, where *Pyr*=[*Pyr*]$\cdot V_{medium}$ (Methods). Thus, in 1 min in the presence of 14 mM [1-^13^C]pyruvate, according to the volumes described in the Methods, one would expect an averaged production of about 0.07 µmole [1-^13^C]lactate. In order to express these rates in a way that will reflect the activity in viable cells, this amount was normalized to the ATP content in each sample (Methods). This resulted in an averaged [1-^13^C]lactate formation rate of 6.1 ± 0.6 nmole/nmole ATP in 1 min.

To eliminate the need for a kinetic model and increase the SNR of [1-^13^C]lactate we have used the approach described in Experimental design 2 (Methods) consisting of a manual bolus injection during perfusion arrest and selective excitations. Typical [1-^13^C]lactate and [1-^13^C]pyruvate signals obtained in this way are shown in Figure 5. Altogether, five injections were recorded in this way from three different samples on three different experimental days. Two samples consisted of tumor slices from three different mice and one sample consisted of tumor slices obtained from an individual mouse (Appendix A). Using the same assumptions with regard to the tissue and the medium volumes (Methods), the rate of [1-^13^C]lactate production was found to range between 0.82 to 14.1 nmole/min/nmole ATP and averaged at 3.9 ± 5.7 nmole/min/nmole ATP. This averaged rate is similar to the rate determined using the same perfusion conditions but with the non-selective acquisition approach, as described above. 

To explore the possibility that the arrested perfusion condition limits metabolism, we have used Experimental design 3 (Methods) in which perfusion was not arrested and the slices were continuously perfused throughout the metabolic measurement. This experimental modification necessitated a modification to the acquisition approach and the analysis (Methods). The main difference in the analysis was in the selection of the time points for quantification of the rate of [1-^13^C]lactate production: When using the perfusion arrest mode (Experimental design 2), where the hyperpolarized dissolution was injected in a bolus injection to the NMR tube (Figure 6A), the concentration of hyperpolarized [1-^13^C]pyruvate is constant for the time that the perfusion is stopped and then reduces as the hyperpolarized medium is washed out and is replaced by regular perfusion medium. To identify the time points in which the pyruvate concentration was constant, the [1-^13^C]pyruvate intensities were corrected for the effect of T_1_ and RF pulses (Methods). These corrected intensities showed a constant level and then a wash-out phase; between the constant level and wash-out phase a small rise in the corrected signal was observed, likely due to the inflow of hyperpolarized [1-^13^C]pyruvate left in the perfusion line which did not experience RF pulsation and that was in a lower magnetic field, hence experienced less loss of polarization than the hyperpolarized [1-^13^C]pyruvate in the probe (Figure 6A). For acquisition utilizing Experimental design 3, a relatively large volume of hyperpolarized medium continuously perfused the tissue slices, and thus the corrected intensities showed: (i) a build-up of the hyperpolarized [1-^13^C]pyruvate concentration; (ii) a constant concentration of hyperpolarized [1-^13^C]pyruvate (steady-state); and (iii) a wash-out phase (Figure 6B). 

The temporal window in both setups in which the concentration of [1-^13^C]pyruvate was constant, was defined as the duration consisting of the time points at which the corrected intensities were within 10% of the first peak value (blue color full circles in Figure 6). Only these time points were used for further analysis of metabolite production rates. In the perfusion arrest system (Figure 6A) this constant phase starts immediately after the hyperpolarization injection and continues until the perfusion is restarted. In the continuous perfusion system (Figure 6B) this constant phase starts approximately 50 s after the first detection of hyperpolarized [1-^13^C]pyruvate signal, and continues until all of the hyperpolarized [1-^13^C]pyruvate containing medium has gone through the NMR tube and non-hyperpolarized DMEM medium starts flowing through the slices, a time frame determined by the total hyperpolarized media volume and the perfusion rate.

After selecting the time points appropriate for quantification of the lactate production rate, the corresponding data points of lactate signal were selected for analysis (Figure 7). For experimental design 3, altogether three injections were performed in this way on one sample on a single experimental day containing tumor slices from one animal. Using the same assumption with regard to the tissue and the medium volumes (Methods), the rate of [1-^13^C]lactate production in Experimental design 3 was found to range between 1.3 to 8.7 nmole/min/nmole ATP (3.8 ± 4.2 nmole/min/nmole ATP). This averaged rate is similar to the rate determined using the arrested perfusion conditions with the non-selective or the selective acquisition approach.

## 4. Discussion

To the best of our knowledge, this is the first study to record real-time ongoing metabolism in fresh, live, breast cancer tissue ex vivo. Tissue preparation encountered technical challenges, due to the rigid nature of the tumor. However, overcoming this obstacle, thin precision-cut tissue slices were adequately produced. Our results document proof of tumor viability for more than 7 h in the spectrometer with a stable ATP level. Furthermore, we have succeeded in repeated and reproducible real-time determination of in-cell LDH enzymatic activity visualized via hyperpolarized [1-^13^C]lactate production. 

The three study set-ups reflect the development of our strategy to quantify the enzymatic activity per viable tissue, avoiding potential confounders on accuracy of measurement, and a future aim for increasing the robustness of this technique. In the first set-up, arresting the perfusion served to limit the medium wash in and wash out on the metabolism. The second set-up was designed to measure only newly produced [1-^13^C]lactate on every excitation by nulling the [1-^13^C]lactate signal produced previously, allowing metabolic rates to be determined without fitting the signals to a kinetic model. The third setup used continuous, undisturbed perfusion flow, which simulates most closely the in vivo conditions. Taking the first study design result as an example, while using a conversion factor for ATP amount to tissue weight of 0.86 μmole/g tissue [61], one may derive LDH activity in a mass to be ~5.2 μmole/g/min or 5.2 mM in 1 min. This measurement may serve as a prediction for LDH activity in hormone responsive breast cancer masses. 

In an attempt to predict the ability to detect such a rate of production of hyperpolarized [1-^13^C]lactate in the human body we will compare this potential signal to the signal of water protons. The concentration of water protons in water is 110 M, therefore 110 M is the upper limit for the concentration of water in body tissues. If we consider a [1-^13^C]lactate concentration of 5.2 mM, following the enhancement of the dDNP process, the signal of this lactate concentration is expected to increase 10,000-fold, therefore the initial hyperpolarized signal of [1-^13^C]lactate can be equivalent to a signal of 52 M of [1-^13^C]lactate. Because the water signal is readily seen on MRI, we predict that a signal of about half the intensity will be visible just as well. In this simplified calculation we have not taken into account issues of sensitivity of ^13^C vs. ^1^H detection or the gradient power required to achieve the image resolution for ^13^C vs. ^1^H. We note that although the sensitivity for ^13^C detection is lower, the potential resolution achieved with the same gradient power is also lower. Indeed, clinical studies performed with hyperpolarized [1-^13^C]pyruvate report a nominal voxel size of 7 × 7 × 10 mm^3^ [20,21]. In such large voxel sizes we predict that detectability of hyperpolarized [1-^13^C]lactate produced in breast cancer tissue will be feasible. As with all other metabolic monitoring of hyperpolarized substrate metabolism the con of the technology is that the magnetization is non-renewable and therefore requires specialized pulse sequences to optimally detect the hyperpolarized metabolites before their signal decays. However, the pro of the technology is that the background signal is expected to be low, as it is highly likely that the cancerous tissues will show higher levels of lactate production, as described in the introduction.

In the current study we have quantified LDH activity in breast cancer and evaluated the potential ability to detect this activity in the human breast using hyperpolarized MR as means for breast cancer detection. However, another important topic is the possible utility of this approach to detect response to treatment. Several in vivo preclinical studies have addressed this issue. For example: Bohndiek et al. [31] demonstrated the early effect of anti VEGF therapy in decreasing the ^13^C flux between hyperpolarized [1-^13^C]pyruvate and [1-^13^C]lactate in the therapy sensitive human colorectal tumor type. Day et al. [34] showed hyperpolarized [1-^13^C]pyruvate-lactate exchange as an early marker for response to treatment in lymphoma bearing mice, as the metabolic pathway was inhibited within 24 h of chemotherapy. However, Asghar Butt et al. [35] compared the k rates of LDH activity in breast cancer tumors before and after treatment with the anti-estrogen drug (tamoxifen), and did not observe a significant change in the reaction rate in the treated group. It remains to be seen whether the effects of other treatments of breast cancer or other breast cancer types will show a metabolic response that can be determined using hyperpolarized [1-^13^C]pyruvate. That monitoring of metabolism can be useful for assessing early response to treatment can be seen in PET studies of breast cancer showing the utility of various metabolic tracers such as ^11^C-choline or [^18^F]fluorothymidine (FLT) [62,63,64]. Ex vivo studies may provide the means to choose the appropriate therapeutic treatment by analysis of the metabolic response of patient derived tumor slices to various therapeutic treatments. The experimental system and acquisition approach developed here can support such studies in the future.

## Figures and Tables

**Figure 1 sensors-19-02089-f001:**
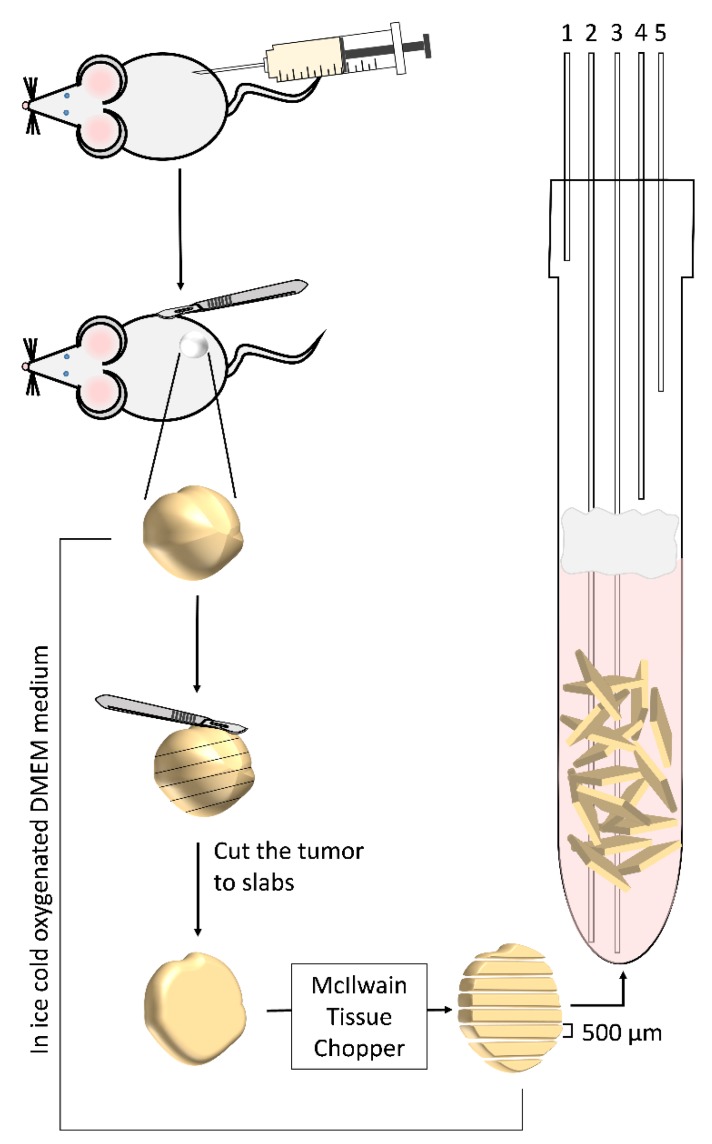
Obtaining xenograft tumors in mice, production of precision cut tumor slices, and maintenance of the slices’ viability in an NMR tube. MCF7 human breast cancer cells ((1–4) × 10^6^) were injected subcutaneously to an immune deficient female mouse. When the tumor reached the size of 1 cm^3^, the tumor was resected (under anesthesia) and placed in ice-cold medium. To cope with the rigidity of the tumor, in order to produce precision-cut tissue slices, the tumor was first cut into *ca.* 1–2 mm slabs, and then each slab was sliced to 500 µm slices and placed in a 10 mm NMR tube with perfusion of DMEM medium at 37 °C at a rate of 4 ml/min. The lines going in and out of the NMR tube are marked as follows: 1. Humidified 95%/5% O_2_/CO_2_ atmosphere; 2. NMR compatible temperature probe; 3. Medium inflow line; 4. Medium out-flow line; 5. Backup outflow line. A filter made of cotton balls (colored gray in the figure) was placed a few cm above the slices to prevent suction of the occasional floating slice into the out-flow lines. The slices were kept at the bottom of the tube at the region visible to the NMR probe due to gravity and possible inter slice adhesion and the gentle perfusion generally did not displace them.

**Figure 2 sensors-19-02089-f002:**
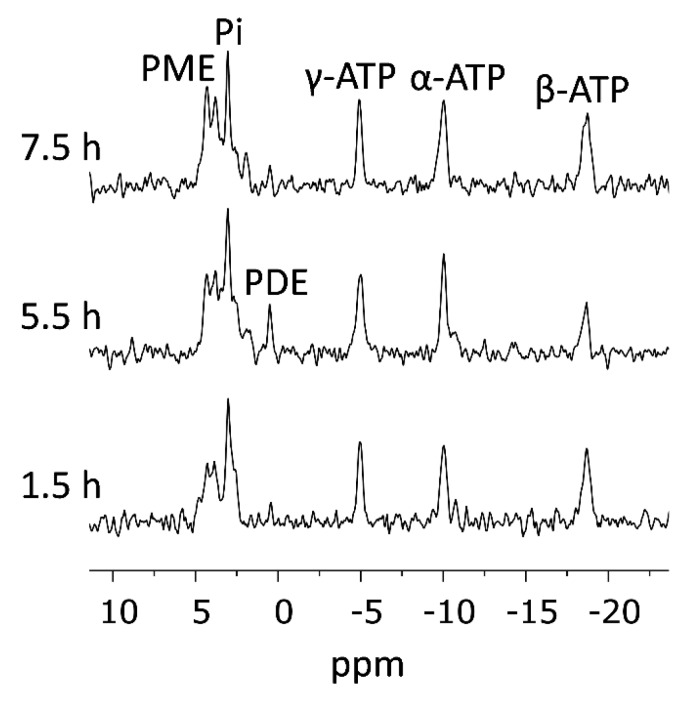
^31^P-NMR spectra acquired from viable breast cancer tumor slices continuously perfused in the NMR spectrometer. Acquisition of the first spectrum started after a recovery period of 0.5 h in the NMR tube, in the spectrometer. The time scale marks the end of the acquisition of each spectrum with reference to the time the slices were placed in the NMR tube. The spectra were acquired with a 50° nutation angle, a repetition time of 1.12 s, and 8192 points. Each spectrum is the sum of two 34 min acquisitions (3600 excitations in total). The spectra were processed with a line-broadening of 15 Hz, zero-filled to 16,384 points, and baseline corrected. PME, phosphomonoesters, typically consisting of phospho-ethanolamine and phosphocholine; Pi, inorganic phosphate; PDE, phosphodiesters; ATP, adenosine triphosphate. The chemical shift was referenced to α-ATP at -10.03 ppm. In this particular example, between 1.5 and 4 h there were 2 injections of hyperpolarized pyruvate (a total of 1.07 mM were added to the perfusion medium) and between 5.5 and 6.5 h the perfusion medium was enriched with choline chloride (1.06 mM), which is likely the reason for the increase in the PME signal in the spectrum acquired at 7.5 h.

**Figure 3 sensors-19-02089-f003:**
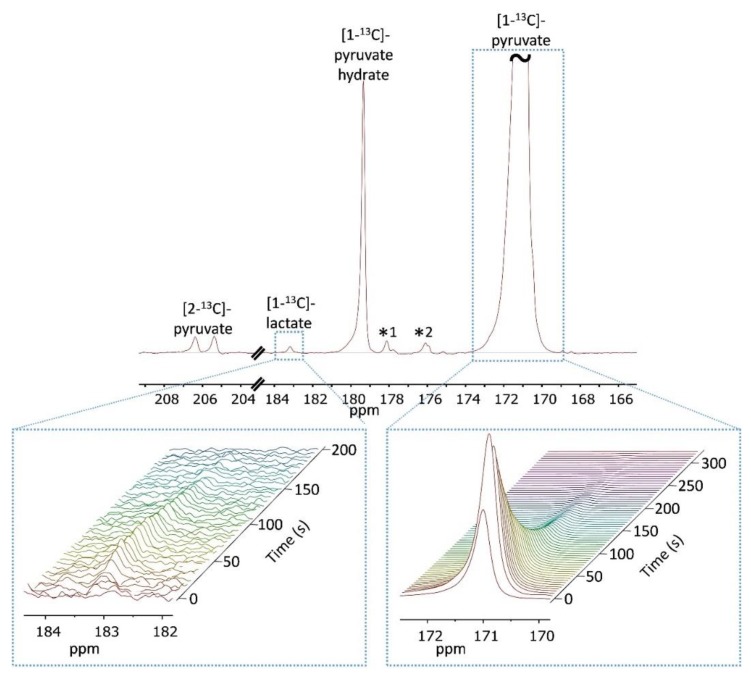
^13^C-NMR spectra of tumor slices in the presence of 14.4 mM hyperpolarized [1-^13^C]pyruvate. The upper panel presents a sum of 16 spectra acquired with a 12° nutation angle, a repetition time of 5 s, and 32,768 points. The spectra were processed with a 5 Hz exponential multiplication and zero filling of 65,536 points. The signal of [1-^13^C]pyruvate was truncated to allow dynamic range for visibility of the metabolite signal. The non-enriched C_2_ position of [1-^13^C]pyruvate appears at 206 ppm. The asterisks (*1 and *2) represent impurities of [1-^13^C]pyruvate, most likely these are due to the C_5_ and C_1_ of zymonic acid (enol) respectively [56,59,60]. The bottom right panel shows the [1-^13^C]pyruvate signal over 330 s. In the first spectrum, the injection is still ongoing and from the 2^nd^ spectrum on, this signal shows only decay. The bottom left panel shows stacked spectra demonstrating a buildup and then a decay of the [1-^13^C]lactate signal over 205 s. The spectra were referenced to C_1_ of pyruvate at 171.0 ppm.

**Figure 4 sensors-19-02089-f004:**
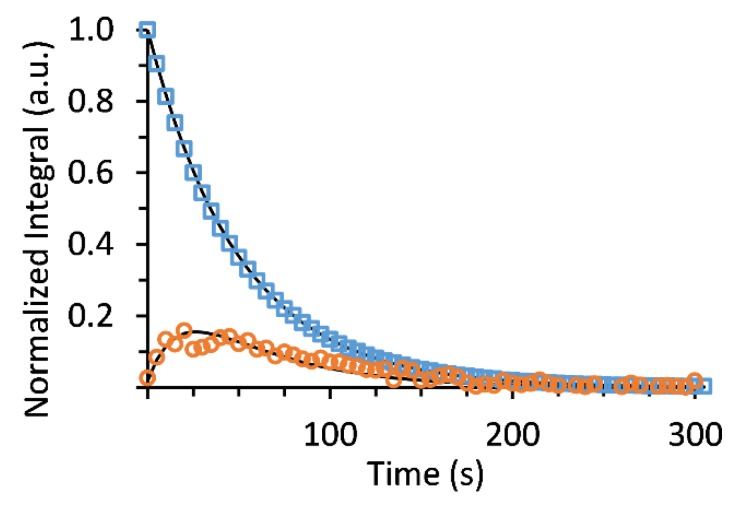
Kinetic model fit of the hyperpolarized signals of [1-^13^C]pyruvate and [1-^13^C]lactate obtained by non-selective RF excitation in the arrested perfusion system. The signals of hyperpolarized [1-^13^C]pyruvate and [1-^13^C]lactate from the same experiment that is presented in Figure 3 were normalized to the highest [1-^13^C]pyruvate signal (on the 2^nd^ spectrum that showed the hyperpolarized [1-^13^C]pyruvate signal). The [1-^13^C]lactate signal is shown multiplied 500-fold. In this example, the rate constant, *k_(1st order)_*, was determined to be $1.55\times 10^{-4}$ s^−1^, the T_1_ for [1-^13^C]pyruvate was 64.5 s, and the T_1_ for [1-^13^C]lactate was 20.0 s. The R^2^ for the [1-^13^C]pyruvate and [1-^13^C]lactate fits were 0.999 and 0.903, respectively.

**Figure 5 sensors-19-02089-f005:**
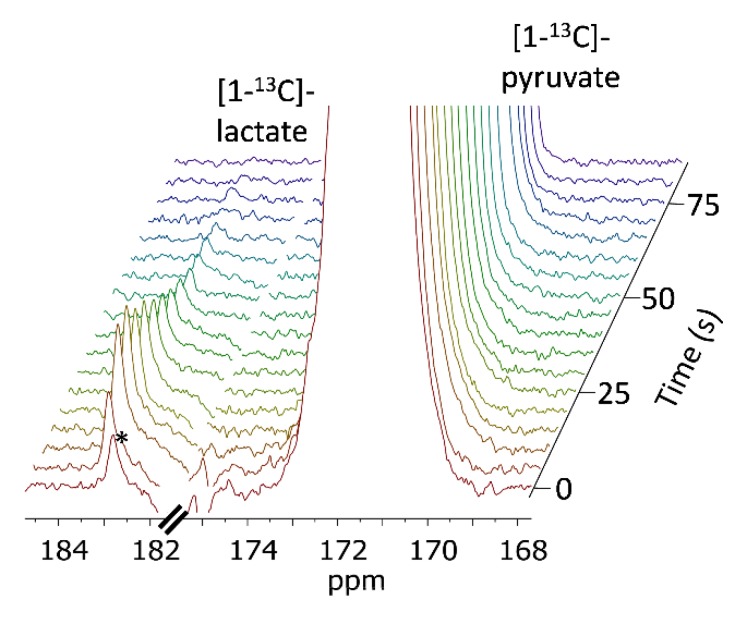
^13^C-NMR spectra of viable MCF7 tissue slices in the presence of 14 mM hyperpolarized [1-^13^C]pyruvate obtained using selective excitations. The spectra were acquired with a repetition time of 5 s and 16,384 points and processed using a 3 Hz exponential multiplication and zero filling to 32,768 points. During 85 s, the [1-^13^C]pyruvate signal is decaying while the [1-^13^C]lactate signal is first increasing and then decays. The signal marked with an asterisk is due to an impurity of [1-^13^C]pyruvic acid [56]. On the 2^nd^ spectrum this signal is depolarized due to the application of the previous selective pulse.

**Figure 6 sensors-19-02089-f006:**
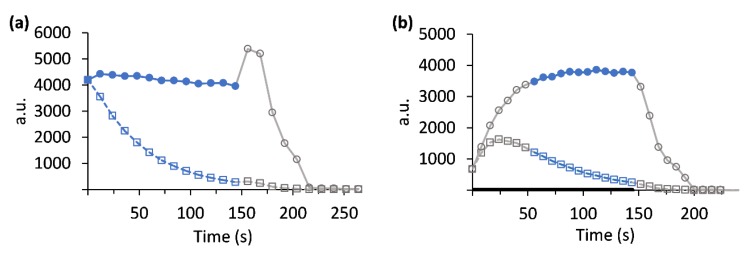
The hyperpolarized signal of [1-^13^C]pyruvate in arrested perfusion and continuous perfusion operation modes (Experimental designs 2 and 3, respectively, Methods) acquired with selective excitations. The hyperpolarized [1-^13^C]pyruvate signal (squares) and the corrected intensities (circles) are presented in arbitrary units for the experimental designs of perfusion arrest (**a**) and continuous perfusion (**b**). The points selected for analysis of the enzymatic rate are marked in blue full circles. The duration in which the hyperpolarized media was flowing through the bypass line (methods) is marked by a thick black line in (**b**).

**Figure 7 sensors-19-02089-f007:**
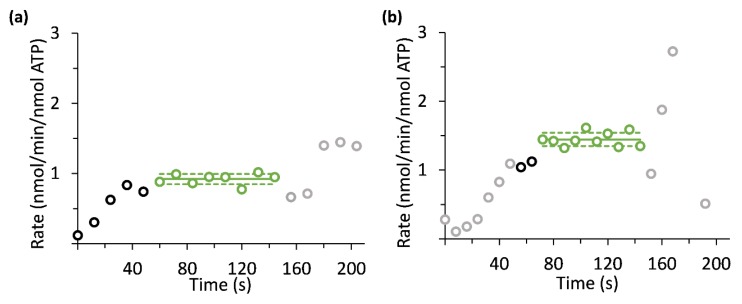
[1-^13^C]lactate production rates per ATP content as determined in live tumor slices using selective excitations in a perfusion arrest and a continuous perfusion experimental designs. One experiment from each experimental design is shown. (**a**) [1-^13^C]lactate production rate as determined in a perfusion arrest system. (**b**) [1-^13^C]lactate production rate as determined in a continuous perfusion system. The dissolution medium was injected at about 27 °C and 37 °C, in (**a**) and (**b**), respectively. The points marked in green and black represent data that were acquired under the condition of constant [1-^13^C]pyruvate concentration. The mean and standard deviation interval of the [1-^13^C]lactate production maximal rate are marked by solid and dashed green lines, respectively. The points in light grey correspond to measurements recorded at a non-constant and therefore unknown concentration of [1-^13^C]pyruvate. Since the amount of [1-^13^C]pyruvate is not known for these points (as they are either in the wash in or in the wash out phase), the rates of production were not quantified at these times. The points in black correspond to measurements recorded at a constant known concentration of [1-^13^C]pyruvate, but before the [1-^13^C]lactate production rate stabilized.

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
