# Peer review of "In-Cell Determination of Lactate Dehydrogenase Activity in a Luminal Breast Cancer Model – ex vivo Investigation of Excised Xenograft Tumor Slices Using dDNP Hyperpolarized [1-13C]pyruvate"

_sensors, 2019, doi:10.3390/s19092089_

Round 1

Reviewer 1 Report

In the manuscript, authors measured LDH activity with [1-13C] pyruvate by using dDNP magnetic resonance in breast cancer. They established a method to measure the LDH activity in fresh tumor specimen by generating precision cut tumor slices. Measuring metabolic state in tumors is one of the important factors for diagnosis. However, studies to measure metabolism with dDNP-MR in breast cancers are currently limited, and this study should provide an useful information in the field.

Following points should be addressed before publication.

1. Authors should measure LDH activity in the cell culture system, and compare with the MCF7 cell-derived tumor samples which were used in this study.

2. What is the expression level of LDH in tumor specimen? It should be quantified in each specimen, and examine if there is any relation to the LDH activity measured with their method.

Author Response

Reviewer #1, general comment #1:

Comments and Suggestions for Authors

In the manuscript, authors measured LDH activity with [1-13C] pyruvate by using dDNP magnetic resonance in breast cancer. They established a method to measure the LDH activity in fresh tumor specimen by generating precision cut tumor slices. Measuring metabolic state in tumors is one of the important factors for diagnosis. However, studies to measure metabolism with dDNP-MR in breast cancers are currently limited, and this study should provide an useful information in the field.

Following points should be addressed before publication.

Authors’ response:

Thank you.

Reviewer #1, comment #1:

1. Authors should measure LDH activity in the cell culture system, and compare with the MCF7 cell-derived tumor samples which were used in this study.

Authors’ response:

Within the time frame allotted for this revision, we will not be able to complete these studies. However, we do agree with the reviewer that this is a limitation of our study. Therefore, we have added this discussion under the limitations section in the Supplementary Materials (S3). In the future, this will be done by monitoring the absorbance at 340 nm which is due to the concurrent production of NADH in the LDH reaction [1].

Reviewer #1, comment #2:

2. What is the expression level of LDH in tumor specimen? It should be quantified in each specimen, and examine if there is any relation to the LDH activity measured with their method.

Authors’ response:

Within the time frame allotted for this revision, we will not be able to complete these studies. However, we do agree with the reviewer that this is a limitation of our study. Therefore, we have added this discussion under the limitations section in the Supplementary Materials (S3). In the future this will be done using multiple assays such as immunohistochemistry (HIC) [1,2], a semi-quantitative analysis of LDH immunolocalization [3], ELISA [3] and western blotting (WB) [4-7]. The tissue samples of the reported studies were unfortunately not saved in a manner that will allow any of these assays, but we do agree that future studies should include these investigations. This is also the case for the response to comment #1.

References

1.            Dong, T.; Liu, Z.; Xuan, Q.; Wang, Z.; Ma, W.; Zhang, Q. Tumor LDH-A expression and serum LDH status are two metabolic predictors for triple negative breast cancer brain metastasis. Sci. Rep. 2017, 7, 6069-6069, doi:10.1038/s41598-017-06378-7.

2.            Koukourakis, M.I.; Giatromanolaki, A.; Sivridis, E. Lactate dehydrogenase isoenzymes 1 and 5: Differential expression by neoplastic and stromal cells in non-small cell lung cancer and other epithelial malignant tumors. Tumour Biol. 2003, 24, 199-202, doi:10.1159/000074430.

3.            Chen, Y., Jr.; Mahieu, N.G.; Huang, X.; Singh, M.; Crawford, P.A.; Johnson, S.L.; Gross, R.W.; Schaefer, J.; Patti, G.J. Lactate metabolism is associated with mammalian mitochondria. Nat. Chem. Biol. 2016, 12, 937-943, doi:10.1038/nchembio.2172.

4.            Javed, M.H.; Azimuddin, S.M.I.; Hussain, A.N.; Ahmed, A.; Ishaq, M. Purification and characterization of lactate dehydrogenase from Varanus liver. Exp. Mol. Med. 1997, 29, 25, doi:10.1038/emm.1997.4.

5.            Fantin, V.R.; St-Pierre, J.; Leder, P. Attenuation of LDH-A expression uncovers a link between glycolysis, mitochondrial physiology, and tumor maintenance. Cancer Cell 2006, 9, 425-434, doi:10.1016/j.ccr.2006.04.023.

6.            Huang, X.; Li, X.; Xie, X.; Ye, F.; Chen, B.; Song, C.; Tang, H.; Xie, X. High expressions of LDHA and AMPK as prognostic biomarkers for breast cancer. Breast 2016, 30, 39-46, doi:10.1016/j.breast.2016.08.014.

7.            Lodi, A.; Woods, S.M.; Ronen, S.M. Treatment with the MEK inhibitor U0126 induces decreased hyperpolarized pyruvate to lactate conversion in breast, but not prostate, cancer cells. NMR Biomed. 2013, 26, 299-306, doi:10.1002/nbm.2848.

Reviewer 2 Report

The work by Adler-Levy et al reports a technical study of the feasibility of performing hyperpolarized [1-13C] pyruvate in perfused tumour tissue slices from a mouse breast xenograft model of MCF-7 cells. Hyperpolarized 13C studies of perfused tumour slices have been previously performed elsewhere in prostate tumour slices in a similar bioreactor type setup, however the novelty in the current work is the application to breast cancer models which is currently being explored clinically in a number of sites. The perfusion system and overall experimental design is appropriate and adequately performed. This will likely yield a wealth of interesting data relating to therapy response or indeed mechanistic studies of tumour biology. However there are a number of shortcomings of the current paper that should be improved. 

1) For such a system to be of use it is important to know what the reproducibility is, either technical for multiple injections into the same tumour slices or biological reproducibility from different tumours. It would appear that this data was acquired in the current study but not reported. Experimental groups were n=3 in the supplementary material therefore rates measured should be presented either as mean +/- SEM or as individual data points to asses the variability.

2) Normalisation of the data in 13C DNP experiment is critical, especially if the effect of therapy is being studied. In this regard ATP is a reasonable surrogate for tissue mass only if the cells are viable. This is not known in the current study, there could be regions of necrosis which would be a confounding factor. Indeed ATP is only really valid for 100% healthy tissue under control conditions. Both the levels of ATP and the rate of pyruvate lactate exchange could be modulated by a therapy therefore this would seriously confound the measurements and interpretation of the data. Did the authors weigh the tissue in the NMR tube either before or after the experiment to normalise for wet weight? Or a more accurate method might be the normalise for protein content. This should be commented on. 

3) A number of impurity peaks of pyruvate have been reported in the literature. Those in the current work look like zymonic acid arising from aldol condensation of pyruvate, please comment.

4) Statistical analysis was performed with the tools available in Excel. Detail of what statistical analysis should be explicitly stated, what tests were used. Indeed further to comment 1 there is little evidence of any statistical analysis of the data?

5) I'm slightly surprised that the 31P spectrum does not contain a peak from phosphocholine, please comment. Given that PC is usually a very prominent metabolite in tumour tissue. Is this expected for MCF7 cells? 

Author Response

Reviewer 2:

Reviewer #2, general comment #1:

Comments and Suggestions for Authors

The work by Adler-Levy et al reports a technical study of the feasibility of performing hyperpolarized [1-13C] pyruvate in perfused tumour tissue slices from a mouse breast xenograft model of MCF-7 cells. Hyperpolarized 13C studies of perfused tumour slices have been previously performed elsewhere in prostate tumour slices in a similar bioreactor type setup, however the novelty in the current work is the application to breast cancer models which is currently being explored clinically in a number of sites. The perfusion system and overall experimental design is appropriate and adequately performed. This will likely yield a wealth of interesting data relating to therapy response or indeed mechanistic studies of tumour biology. However there are a number of shortcomings of the current paper that should be improved.

Authors’ response:

Thank you.

Reviewer #2, comment #1:

For such a system to be of use it is important to know what the reproducibility is, either technical for multiple injections into the same tumour slices or biological reproducibility from different tumours. It would appear that this data was acquired in the current study but not reported. Experimental groups were n=3 in the supplementary material therefore rates measured should be presented either as mean +/- SEM or as individual data points to asses the variability.

Authors’ response:

The rate calculated for each experiment (each injection) was added to the Supplementary Materials in Table S2.

Reviewer #2, comment #2:

Normalisation of the data in 13C DNP experiment is critical, especially if the effect of therapy is being studied. In this regard ATP is a reasonable surrogate for tissue mass only if the cells are viable. This is not known in the current study, there could be regions of necrosis which would be a confounding factor. Indeed ATP is only really valid for 100% healthy tissue under control conditions. Both the levels of ATP and the rate of pyruvate lactate exchange could be modulated by a therapy therefore this would seriously confound the measurements and interpretation of the data. Did the authors weigh the tissue in the NMR tube either before or after the experiment to normalise for wet weight? Or a more accurate method might be the normalise for protein content. This should be commented on.

Authors’ response:

We agree with the reviewer that this is an important point not discussed in the manuscript. We have added this discussion under the limitations section in the Supplementary Materials (S3).

Reviewer #2, comment #3:

A number of impurity peaks of pyruvate have been reported in the literature. Those in the current work look like zymonic acid arising from aldol condensation of pyruvate, please comment.

Authors’ response:

We agree with the reviewer and have added this information to Figure 1 and the legend of Figure 1.

Reviewer #2, comment #4:

Statistical analysis was performed with the tools available in Excel. Detail of what statistical analysis should be explicitly stated, what tests were used. Indeed further to comment 1 there is little evidence of any statistical analysis of the data?

Authors’ response:

Indeed, the only statistical analysis was calculation of standard deviations. We have removed this sentence from the manuscript.

Reviewer #2, comment #5:

5. I'm slightly surprised that the 31P spectrum does not contain a peak from phosphocholine, please comment. Given that PC is usually a very prominent metabolite in tumour tissue. Is this expected for MCF7 cells?

Authors’ response:

The phosphocholine peak in our experiments was marked as phosphomonoesters (PME) in Figure 2. We have now stated this in the legend to Figure 2. Going through many publications on MCF-7 tumors it can be observed that the PME peak in our spectra (which likely contains phosphocholine) is in agreement with previous literature. As a gross parameter for evaluation of the reviewer’s question we have compared the intensity of this signal to the signals of ATP. In the literature, this peak is sometimes higher than that of the ATPs peaks [1-6] and sometimes at the same level of the ATP signals [7-13]. In our study, the PME signal is at similar level to that of the ATP signals. The content of choline or ethanolamine in the medium will affect the size of the PME signal as well.

References

1.            Ward, C.S.; Eriksson, P.; Izquierdo-Garcia, J.L.; Brandes, A.H.; Ronen, S.M. HDAC inhibition induces increased choline uptake and elevated phosphocholine levels in MCF7 breast cancer cells. PloS one 2013, 8, e62610-e62610, doi:10.1371/journal.pone.0062610.

2.            Brandes, A.H.; Ward, C.S.; Ronen, S.M. 17-allyamino-17-demethoxygeldanamycin treatment results in a magnetic resonance spectroscopy-detectable elevation in choline-containing metabolites associated with increased expression of choline transporter SLC44A1 and phospholipase A2. Breast Cancer Res. 2010, 12, R84-R84, doi:10.1186/bcr2729.

3.            Katz-Brull, R.; Margalit, R.; Bendel, P.; Degani, H. Choline metabolism in breast cancer; 2H-, 13C- and 31P-NMR studies of cells and tumors. MAGMA 1998, 6, 44-52, doi:10.1007/BF02662511.

4.            Degani, H.; Furman, E.; Fields, S. Magnetic resonance imaging and spectroscopy of MCF7 human breast cancer: Pathophysiology and monitoring of treatment. Clin. Chim. Acta 1994, 228, 19-33, doi:10.1016/0009-8981(94)90054-X.

5.            Furman, E.; Rushkin, E.; Margalit, R.; Bendel, P.; Degani, H. Tamoxifen induced changes in MCF7 human breast cancer: In vitro and in vivo studies using nuclear magnetic resonance spectroscopy and imaging. J. Steroid Biochem. Mol. Biol. 1992, 43, 189-195, doi:10.1016/0960-0760(92)90207-Y.

6.            Singer, S.; Souza, K.; Thilly, W.G. Pyruvate utilization, phosphocholine and adenosine triphosphate (ATP) are markers of human breast tumor progression: a 31P- and 13C-nuclear magnetic resonance (NMR) spectroscopy study. Cancer Res. 1995, 55, 5140-5145, doi:10.1016/0006-291x(82)90853-1

7.            Raghunand, N.; He, X.; van Sluis, R.; Mahoney, B.; Baggett, B.; Taylor, C.W.; Paine-Murrieta, G.; Roe, D.; Bhujwalla, Z.M.; Gillies, R.J. Enhancement of chemotherapy by manipulation of tumour pH. Br. J. Cancer 1999, 80, 1005-1011, doi:10.1038/sj.bjc.6690455.

8.            Pilatus, U.; Aboagye, E.; Artemov, D.; Mori, N.; Ackerstaff, E.; Bhujwalla, Z.M. Real-time measurements of cellular oxygen consumption, pH, and energy metabolism using nuclear magnetic resonance spectroscopy. Magn. Reson. Med. 2001, 45, 749-755, doi:10.1002/mrm.1102.

9.            Shah, T.; Wildes, F.; Penet, M.-F.; Winnard, P.T., Jr.; Glunde, K.; Artemov, D.; Ackerstaff, E.; Gimi, B.; Kakkad, S.; Raman, V., et al. Choline kinase overexpression increases invasiveness and drug resistance of human breast cancer cells. NMR Biomed. 2010, 23, 633-642, doi:10.1002/nbm.1510.

10.         Raghunand, N.; Altbach, M.I.; van Sluis, R.; Baggett, B.; Taylor, C.W.; Bhujwalla, Z.M.; Gillies, R.J. Plasmalemmal pH-gradients in drug-sensitive and drug-resistant MCF-7 human breast carcinoma xenografts measured by 31P magnetic resonance spectroscopy. Biochem. Pharmacol. 1999, 57, 309-312, doi: 10.1016/S0006-2952(98)00306-2

11.         Viti, V.; Ragona, R.; Guidoni, L.; Barone, P.; Furman, E.; Degani, H. Hormonally induced modulation in the phosphate metabolites of breast cancer: analysis of in vivo 31P MRS signals with a modified prony method. Magn. Reson. Med. 1997, 38, 285-295, doi:10.1002/mrm.1910380219.

12.         Maril, N.; Degani, H.; Rushkin, E.; Sherry, A.D.; Cohn, M. Kinetics of cyclocreatine and Na+ cotransport in human breast cancer cells: mechanism of activity. Am. J. Physiol. 1999, 277, C708-716, doi:10.1152/ajpcell.1999.277.4.C708.

13.         Wijnen, J.P.; Jiang, L.; Greenwood, T.R.; van der Kemp, W.J.M.; Klomp, D.W.J.; Glunde, K. 1H/31P polarization transfer at 9.4 Tesla for improved specificity of detecting phosphomonoesters and phosphodiesters in breast tumor models. PloS one 2014, 9, e102256-e102256, doi:10.1371/journal.pone.0102256.

Round 2

Reviewer 2 Report

The authors have adequately responded to my comments.